# Influence of landscape structure on carbon storage in agroforestry systems with cacao and silvopastoral systems in the Colombian Amazon

Jenniffer Tatiana Díaz-Cháux[1,2]*, Alexander Velasquez-Valencia[1],
Fernando Casanoves[2,3]

1 University of the Amazon, Andean Amazon Biodiversity Research Center, Wildlife Research Group, Faculty of Basic Sciences, Biology Program, Florencia, Caquetá, Colombia, 2 University of the Amazon, Doctoral Program in Natural Sciences and Sustainable Development, Florencia, Caquetá, Colombia, 3 CATIE - Tropical Agricultural Research and Higher Education Center, Turrialba, Costa Rica

* j.diaz@udla.edu.co

## Abstract

In the Colombian Amazon region, agricultural and livestock activities lead to changes in land use, transforming complex and heterogeneous natural landscapes into landscapes characterized by a matrix of pastures and forest fragments with low connectivity. These agroforestry landscapes play a crucial role in biodiversity conservation and the carbon cycle. The objective of this research was to determine the influence of landscape structure and spatial configuration in cacao-based agroforestry systems (SAFc) and silvopastoral systems (SSP) on carbon storage in the Colombian Amazon. The study was conducted across eight mosaics of rural production landscapes in the Amazon region, each consisting of a 1 km² grid where vegetation covers were classified, and landscape metrics were quantified. A total of 78 plots of 0.1 ha were established in 44 cover patches within the SAFc and SSP mosaics, and dasometric variables were measured to inventory carbon deposits in aboveground biomass, root biomass, litter, and herbaceous vegetation. It was estimated that, in the Colombian Amazon, the studied SAFc and SSP systems store an average of 15.20 Mg C ha⁻¹ in their biomass. Carbon storage was positively correlated with landscape aggregation metrics and spatial configuration within the system mosaics. Mosaics with patches of symmetrical shapes and lower irregularity, exhibiting greater contiguity, showed higher biomass and carbon storage. Therefore, productive landscapes with complex and connected mosaics enhance the provision of regulatory ecosystem services through carbon storage. Restoration efforts in fragmented areas should be managed at the landscape level by expanding the area of planting patches, establishing patches with regular geometric configurations, and improving connectivity among patches of the same type.

**Data availability statement:** All supplementary material files are available in the Figshare database (doi: 10.6084/m9.figshare.28792067).

**Funding:** This study was supported by the Programa de Becas de Excelencia Doctoral del Bicentenario through a grant awarded to JTD (BPIN 2021000100036), and by MINCIENCIAS through a salary provided to JTD. The specific contributions of this author are detailed in the 'Author Contributions' section. The funding agencies had no role in the study design, data collection and analysis, decision to publish, or preparation of the manuscript.

**Competing interests:** Los autores han declarado que no existen intereses en conflicto.

## Introduction

The Amazon biome covers 41.82% of Colombian territory and contains 66% of the country's tropical moist forests [1]. Its environmental conditions and ecosystem functions contribute to global climate change adaptation and mitigation [2,3]. Despite its importance, over 1,858,285 hectares of natural forests were deforested in the past decade, representing 65% of Colombia's total deforestation. The primary causes of natural cover loss include the expansion of agricultural and livestock systems, as well as the establishment of illicit crops incompatible with the land's potential [4–6].

These land-use changes create a mosaic of habitats consisting of a heterogeneous mix of vegetation patches at different successional stages, influencing landscape structure, composition, ecosystem patterns, and processes [5,7]. Additionally, they impact forest diversity, reduce carbon storage, and increase greenhouse gas (GHG) emissions [8]. According to Global Forest Watch [9], activities within the AFOLU module (Agriculture, Forestry, and Other Land Use) are responsible for 59% of emissions, totaling 178.75 Gg of $CO_2$e in the Colombian Amazon, thereby increasing vulnerability to climate change. These figures underscore a conflict regarding land tenure and productive land use across much of the Amazon region [10,11].

In this context, agroforestry systems present a productive, economic, and environmental alternative as part of nature-based solutions for climate change adaptation (NBS-CCA) [12]. Managing rural landscapes through these systems helps mitigate the effects of deforestation and forest fragmentation, supporting the integrity and connectivity of habitat patches [13,14]. Likewise, the structural complexity and heterogeneity of mosaics in productive landscapes influence soil restoration, biodiversity distribution, and the provision of ecosystem services such as carbon storage [15], thereby reducing climate change impacts.

Agroforestry systems with cacao (*Theobroma cacao* L.) (SAFc) and silvopastoral systems (SSP) are recognized as significant sinks for atmospheric $CO_2$. According to Díaz-C et al. [3], carbon fixation and storage in biomass, necromass, and soils depend on the vegetation structure used, climatic variables [16–18], and the system's establishment age [19]. Melito et al. [20], suggest that landscape mosaics of productive systems with an agroforestry crop matrix and high connectivity among patches of dense vegetation enhance carbon storage and reduce greenhouse gases like $CO_2$ [19]. Numerous studies have examined carbon storage in various forest components within productive systems across Colombia: Tolima [16,21–25], Meta [18,26–28], Chocó [29,30], Casanare [31], Antioquia [32], Cundinamarca [33], Caquetá [34,35], and the Amazon region [36–40]. However, studies analyzing the relationship between this regulatory ecosystem service and the structure and composition of rural production landscapes remain scarce. This research addresses the knowledge gap by exploring how the structure and composition of agroforestry systems with cacao and silvopastoral systems in the Colombian Amazon influence carbon storage in aboveground biomass.

The objective of this study was to assess the influence of landscape structure and spatial configuration in cacao agroforestry systems and silvopastoral systems on

carbon storage in the Colombian Amazon. The analysis was conducted at the mosaic scale by selecting metrics describing spatial configuration and heterogeneity. Vegetation covers were classified by patch within each mosaic, where temporary plots were established. Dasometric variables were measured to estimate carbon stored in aboveground and root biomass, as well as in herbaceous vegetation and litter. The working hypothesis posits that biomass and carbon stored in the agroforestry landscapes of the Colombian Amazon increase in mosaics with greater heterogeneity of natural covers and higher connectivity among their patches. Thus, mosaics with a greater variety of natural vegetation cover types should provide more regulatory ecosystem services due to their higher biomass content in forest components and greater carbon fixation rates.

## Materials and methods

### Study area

The study was conducted in the northwestern region of the Colombian Amazon, across seven municipalities in the Caquetá department: Albania (01º19'N, 75º52'W, 277 m a.s.l.), Doncello (01º40'N, 75º16'W, 359 m a.s.l.), Florencia (01º36'N, 75º36'W, 272 m a.s.l.), Montañita (01º36'N, 75º36'W, 269 m a.s.l.), Milán (01º17'N, 75º30'W, 222 m a.s.l.), Morelia (01º29'N, 75º43'W, 259 m a.s.l.), and San José del Fragua (01º19'N, 75º58'W, 376 m a.s.l.) (Fig 1). The region is characterized by dominant geomorphology of hills, foothills, and floodplain valleys with slopes less than 12% [7]. Annual precipitation totals 4 277 mm, with a unimodal distribution and peak rainfall between April and October. The average temperature is 28.62 °C, and relative humidity is 86%.

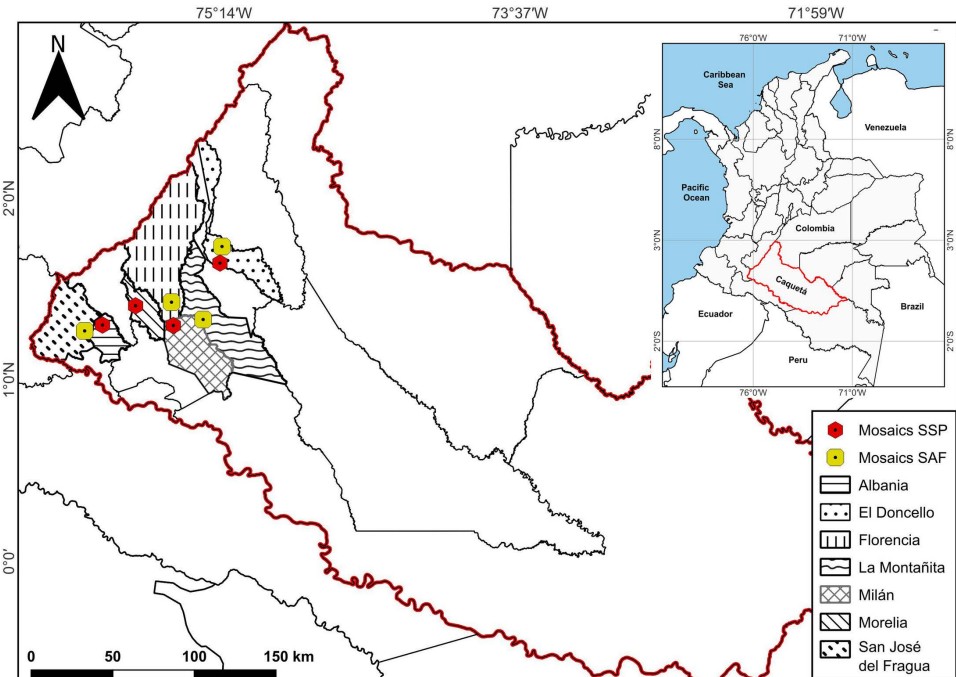

**Fig 1. Location of landscape mosaics with agroforestry and silvopastoral systems in the Colombian Amazon.** SAFc: TR (El Triunfo – Doncello), BA (Batalla 13 – Florencia), SR (Santa Rosa – San José del Fragua), TE (El Tesoro – Montañita). SSP: PO (El Porvenir – Albania), VE (La Vega – Doncello), VM (Villa Mery – Morelia), ES (Esmeraldas – Milán). Source: The map was developed by the authors using QGIS Version 3.40.0, the map was cross verified with the Colombia map, including Departments and Territories' boundaries as shown in the official website of IGAC (public domain) of Colombia: https://geoportal.igac.gov.co.

In the study area, two agroforestry production systems were characterized: cacao agroforestry systems (SAFc) and silvopastoral systems (SSP), both pivotal to the Amazon region's economy [41]. In the Caquetá department, agriculture contributes 18.49% to the Gross Domestic Product (GDP) [42]. Within this sector, livestock accounts for 8.51% of the departmental GDP, supporting approximately 14 000 families across 21 070 farms, with a cattle population of 2 293 751 [43]. The carbon footprint is 19.6 kg $CO_2$e per kilogram of beef and 1.63 kg $CO_2$e per liter of milk [44]. Cacao cultivated under the agroforestry system model produces 393.8 kg ha$^{-1}$ and contributes 0.70% to the department's agricultural GDP, covering 4 488 hectares and benefiting around 1 200 cacao-farming families [45,46].

### Methods analysis of landscape structure and configuration with agroforestry systems with cocoa and silvopastoral systems

The study spanned eight landscape mosaics with rural production systems: four in cacao agroforestry systems (SAFc) and four in silvopastoral systems (SSP). Each mosaic consisted of a 1 km² (100 ha) grid that allowed for the inclusion of multiple types of vegetation cover and landscape elements [7,19]. The classification of vegetation covers in the mosaics was performed using digital processing of Landsat TM satellite images with a resolution of 30 m, employing QGis 3.36 software and based on the guidelines proposed in the CORINE Land Cover methodology adapted for Colombia [47]. Validation and verification of the vegetation cover map for each mosaic were completed during field outings conducted in the primary data collection period (Fig 2).

The studied cacao-based agroforestry systems are characterized by high structural and floristic diversity, integrating both timber-yielding tree species and others of productive and commercial interest. Among the most common timber species are *Parkia velutina*, *Eschweilera albiflora*, *Croton lechleri*, *Conceveiba pleiostemona*, and *Terminalia amazonia*, which contribute to canopy cover, microclimatic regulation, and provide essential ecological functions such as habitat for local fauna. These species coexist with economically important cultivated plants such as *Theobroma cacao*, *Hevea brasiliensis*, *Citrus limon*, *C. reticulata*, *Inga edulis*, *Cedrela odorata*, and *Cordia alliodora*, forming a multi-strata system that promotes both ecological and productive sustainability.

In contrast, silvopastoral systems exhibit a more simplified vegetation structure and lower floristic diversity. Their composition combines native species resulting from natural regeneration processes—such as *Psidium guajava*, *Samanea*

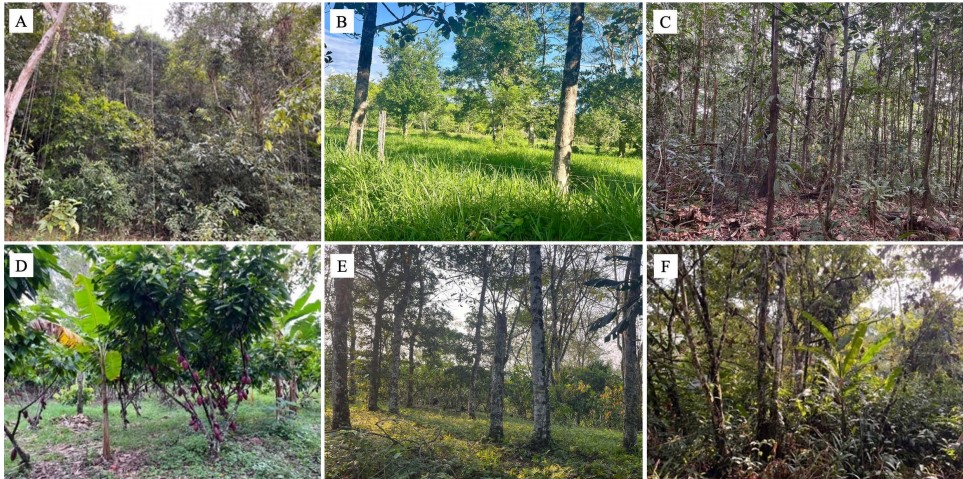

**Fig 2. Vegetation cover types classified within landscape mosaics of cocoa-based agroforestry systems and silvopastoral systems in the Colombian Amazon.** (A) secondary forest – BSE; (B) pastures with scattered trees – PAD; (C) early successional fallows – RTT; (D) cocoa agroforestry crops – CA; (E) shade trees for cocoa crops; (F) late successional fallows – RV.

*saman*, and *Zygia longifolia*—with introduced species established through planting and restoration practices aimed at enhancing ecological connectivity, vegetation cover, and shade availability for livestock. These include forage and nitrogen-fixing species such as *Acacia mangium*, *Anadenanthera peregrina*, *Syzygium cumini*, and *Gmelina arborea*, which also play a functional role in soil recovery and the enhancement of livestock productivity within a sustainable landscape management framework.

The analysis of the landscape structure for each mosaic was conducted following the methodology proposed by Velasquez-Valencia & Bonilla-Gómez [7]. The selected metrics included variables describing the composition and configuration of the landscape mosaics and the potential explanatory power of each metric for carbon sequestration (Table 1) [19,48,49]. Metric calculations were performed using FragStat software version 4.2 [50] based on the classified vegetation cover maps for each mosaic.

## Quantification of biomass in carbon deposits

Between January and November 2023, 78 temporary biomass and carbon plots of 0.1 hectares (20 x 30 m) [51] were established: 40 in SAFc across five vegetation cover types and 38 in SSP across four cover types. The plots were systematically placed in the classified vegetation covers within each mosaic, where samples were collected to inventory carbon in aboveground biomass, root biomass, litter, and herbaceous vegetation. The cover type with the highest number of plots was early fallow (RTT), followed by old fallow (RTV) and cacao agroforestry (CAC).

In each plot, the diameter at breast height (dbh) of all trees (dbh ≥ 10 cm) was measured, along with the diameter at 30 cm above ground (d30) for cacao shrubs [30,51]. Across both systems, a total of 2 846 individual plants were surveyed: 1 341 trees in SSP mosaics and 1 505 individuals in SAFc mosaics, including 330 cacao shrubs, 105 *Hevea brasiliensis* trees, 29 palms, thirteen fruit trees, and 1 028 saplings.

Aboveground biomass was calculated using allometric models developed for the specific life zone and plant species encountered (Table 2). Root biomass was estimated from aboveground biomass values using the model proposed by Cairns et al. [52], as recommended by the Intergovernmental Panel on Climate Change [53].

To calculate biomass in litter and herbaceous vegetation, samples were collected using 0.25 m² frames systematically distributed across each plot. Five frames were installed for herbaceous plants and three for litter per plot. In total, 390

**Table 1. Metrics for landscape properties analysis. Analysis of composition, structure, and spatial configuration of mosaics with agroforestry and silvopastoral systems in the Amazon, Colombia.**

| Landscape property | Variable | Initials | Description |
|---|---|---|---|
| Heterogeneity | Patches number | NP | Measures the fragmentation of the landscape |
| | Total area | CA | Total area (hectares) of patches of the same type and total area of the landscape |
| | Average Fractal Index | FRAC | Measures the complexity of patch edges by the ratio of the logarithms of their perimeters and areas. |
| Aggregation | Distance to nearest neighbor | ENN | Distance to the nearest fragment of the same class. |
| | Contiguity index | CONTIG | Orthogonal and diagonal adjacency of the pixels in each patch |
| Space configuration | Average shape index | SHAPE | Measures the complexity of the form |
| | Area | AREA | Calculate the area of each of the fragments |
| | Turning radius | GYRATE | Distance of a pixel from the centroid of the patch. |
| | Perimeter-Area Ratio | PERIM | Divide the perimeter of a patch by the area. |
| | Average perimeter/area relation | PARA | Measure the ratio of the perimeter to the area of each patch. |

The average values of these landscape metrics per vegetation cover type across the classified mosaics are presented in Table S1 of the supplementary material

**Table 2. Allometric models for estimating aboveground biomass in temporary plots within the landscape mosaics with agroforestry and silvopastoral systems in the Amazon, Colombia.**

| Allometric model | Description | Author |
|---|---|---|
| $AGB = \exp(2.406 - 1.289 \cdot \ln(dbh) + 1.169 \cdot \ln(dbh)^2 - 0.122 \cdot \ln(dbh)^3 + 0.445 \cdot \ln(D))$ | Trees (multispecies) | Álvarez et al. [54] |
| $\ln(AGB) = -3.74 + 2.63 \cdot \ln(d_{30})$ | Cacao shrubs (*Theobroma cacao*) | Andrade et al. [55] |
| $\ln(AGB) = -2.57 + 2.65 \cdot \ln(dbh)$ | Fruit trees | Andrade et al. [55] |
| $\ln(AGB) = -2.99 + 2.72 \cdot \ln(dbh)$ | Rubber trees (*Hevea brasiliensis*) | Andrade et al. [56] |
| $\ln(AGB) = -3.3488 + 2.7483 \cdot \ln(dbh)$ | Palms (Arecaceae) | Goodman et al. [57] |
| $Br = \exp^{(-1.0587 + 0.8836 \cdot \ln(AGB))}$ | Root biomass | Cairns et al. [52] IPCC [53] |

AGB = above-ground biomass (kg tree$^{-1}$); D = wood density (g m$^{-3}$); dbh = diameter at breast height (cm); d30 = trunk diameter at 30 cm from the ground (cm); Br = root biomass (Mg ha$^{-1}$). A wood density value of 0.69 g cm$^{-3}$, recommended for tropical regions, was used [54,58].

herbaceous vegetation subplots were sampled (200 in SAFc and 190 in SSP), along with 234 litter subplots (120 in SAFc and 114 in SSP). All material from each sample was collected and weighed to obtain the wet weight, and a 250 g subsample from each frame was dried at 65 °C in the laboratory to estimate the percentage of dry matter [59]. Using the wet and dry weights, the percentage of dry matter (%DM) was calculated to determine the dry biomass in litter and herbaceous vegetation [51]. The dasometric data matrix per plot established in each of the eight mosaics of the sampled systems are presented in Table S2 of the supplementary material.

## Data analysis

Based on the total biomass estimated per plot—calculated as the sum of aboveground biomass, root biomass, herbaceous vegetation, and litter, expressed in Mg ha$^{-1}$ the total carbon stored in each component (aboveground, roots, herbaceous vegetation, and litter) was determined. A carbon fraction of 0.47 was applied for total carbon estimation [3,58]. These carbon stock values per hectare were then extrapolated to the total area of each vegetation cover type, as well as to the total area of the mosaics within the two productive systems studied: cacao-based agroforestry systems (SAFc) and silvopastoral systems (SSP). Additionally, the carbon sequestration rate was analyzed in terms of $CO_2$ equivalents, using the stoichiometric conversion factor of 3.67, which represents the molecular weight ratio of $CO_2$ to C [22,58]. Results were expressed in gigagrams (Gg), and statistical analyses were performed using *InfoStat* software, version 2020 [60].

To estimate differences in the mean patch area of vegetation cover types among the eight landscape mosaics studied under cacao-based agroforestry systems and silvopastoral systems, an analysis of variance (ANOVA) was conducted. The normality of the variables was tested using the Shapiro–Wilk test, and homoscedasticity was assessed with the F-test. A post-ANOVA pairwise mean comparison was performed using Fisher's LSD method ($\alpha = 0.05$) to confirm differences among vegetation cover types. To analyze the relationship between the metrics describing the structure and spatial configuration of the landscape, the eight studied mosaics, and the five classified vegetation cover types, a Principal Component Analysis (PCA) was conducted to produce a bi-plot. This analysis was performed separately for the landscape metrics, using the eight mosaics and the five classified vegetation cover types, employing the *InfoStat* statistical software, version 2020 [60].

To assess differences between the agroforestry and silvopastoral systems, as well as among mosaics and vegetation cover types, in relation to biomass production and carbon storage in the various components, an analysis of variance was performed using Generalized Linear Mixed Models (GLMMs). The dependent variable was carbon content in each component: aboveground biomass, herbaceous vegetation, litter, and total carbon. Mosaics and vegetation cover types were considered fixed effects. Mean comparisons were conducted using Fisher's LSD test ($\alpha = 0.05$), with statistical analyses performed in *InfoStat* version 2020 [60].

To explore the relationship between carbon stored in each component (aboveground, herbaceous, litter, and total carbon), landscape mosaics, vegetation cover types, and landscape structure and configuration metrics, a Partial Least Squares (PLS) regression was performed to produce a tri-plot. This analysis was also conducted separately, relating landscape metrics and stored carbon to the mosaics and vegetation cover types, using *InfoStat* software version 2020 [60].

## Results

### Analysis of landscape structure and configuration with agroforestry systems with cocoa and silvopastoral systems

A total of five types of vegetation cover were identified in the landscapes with production systems. The largest cover types by area were early fallow (RTT), accounting for 47.62% and present in all mosaics, and pastures with scattered trees (PAD), covering 23.00%. The average area of cover types across the mosaics showed significant differences for BSE (F = 163.76; p = 0.0001; df = 7), RTT (F = 6.98; p = 0.0001; df = 34), RTV (F = 8.97; p = 0.0013; df = 16), and CAC (F = 160.05; p = 0.0001; df = 11). The latter cover type (CAC) was present only in SAFc mosaics (Table 3).

In the principal component analysis (PCA) of landscape metrics for cacao agroforestry systems (SAFc) and silvopastoral systems (SSP), the first two components explained 79.6% of the total variability (Fig 3A). The first component grouped, on the positive end, mosaics characterized by greater patch shape complexity. On the negative end, it associated mosaics with smaller, more aggregated patches. The second component grouped mosaics with irregular and highly aggregated patches on the positive axis, whereas the negative end represented mosaics with larger patches exhibiting smoother and less irregular contours.

In the PCA of vegetation cover metrics, the components explained 91.7% of the variability in the data (Fig 3B). The first component grouped, on the positive axis, cover types with larger patches that were irregular in shape and less compact. On the negative axis, it associated cover types with smaller, more regularly shaped patches that were more spatially isolated. This component effectively differentiates between extensive open pasture covers and early successional fallow vegetation. The second component, on its positive end, grouped cover types with patches exhibiting complex and irregular shapes. On the negative axis, it reflected an association with cover types characterized by higher tree density, larger patch sizes, and greater spatial aggregation

**Table 3. Mean area (hectares) and standard deviation (hectares) of the vegetation covers types classified in the landscape mosaics with agroforestry and silvopastoral systems in the Amazon, Colombia.** Values sharing a common letter in their means are not significantly different (p > 0.05). Pairwise means comparison method: Fisher, alpha = 0.05.

| Systems | Mosaics | BSE | | CAC | | PAD | | RTT | | RTV | |
|---|---|---|---|---|---|---|---|---|---|---|---|
| | | Area | SD± | Area | SD± | Area | SD± | Area | SD± | Area | SD± |
| SAFc | BA | | | 5.57 B | 0.00 | 15.39 A | 0.00 | 8.83 A | 1.95 | 23.64 C | 0.00 |
| | SR | 1.96 A | 0.00 | 5.18 A | 1.37 | | | 0.38 A | 0.00 | 17.21 AB | 10.67 |
| | TE | | | 6.93 A | 0.09 | | | 55.71 A | 36.16 | 9.62 A | 0.67 |
| | TR | | | 10.83 C | 0.00 | 67.68 A | 39.63 | 3.53 A | 0.00 | | |
| SSP | ES | | | | | 16.84 A | 0.00 | 55.03 A | 7.45 | 17.13 BC | 0.00 |
| | PO | | | | | 19.52 A | 0.00 | 37.93 A | 4.38 | 11.94 A | 1.30 |
| | VE | 8.14 A | 2.59 | | | | | 44.69 B | 0.00 | 16.76 AB | 1.50 |
| | VM | 17.77 B | 0.00 | | | | | 41.34 A | 14.55 | | |

SAFc: cacao-based agroforestry systems; SSP: silvopastoral systems; BA: Batalla 13; TE: El Tesoro; TR: El Triunfo; SR: Santa Rosa; ES: Esmeraldas; PO: El Porvenir; VE: La Vega; VM: Villa Mery; BSE: Secondary forest; CAC: Agroforestry cacao cultivation; PAD: Pastures with scattered trees; RTT: Early Fallows; RTV: Old Fallows.

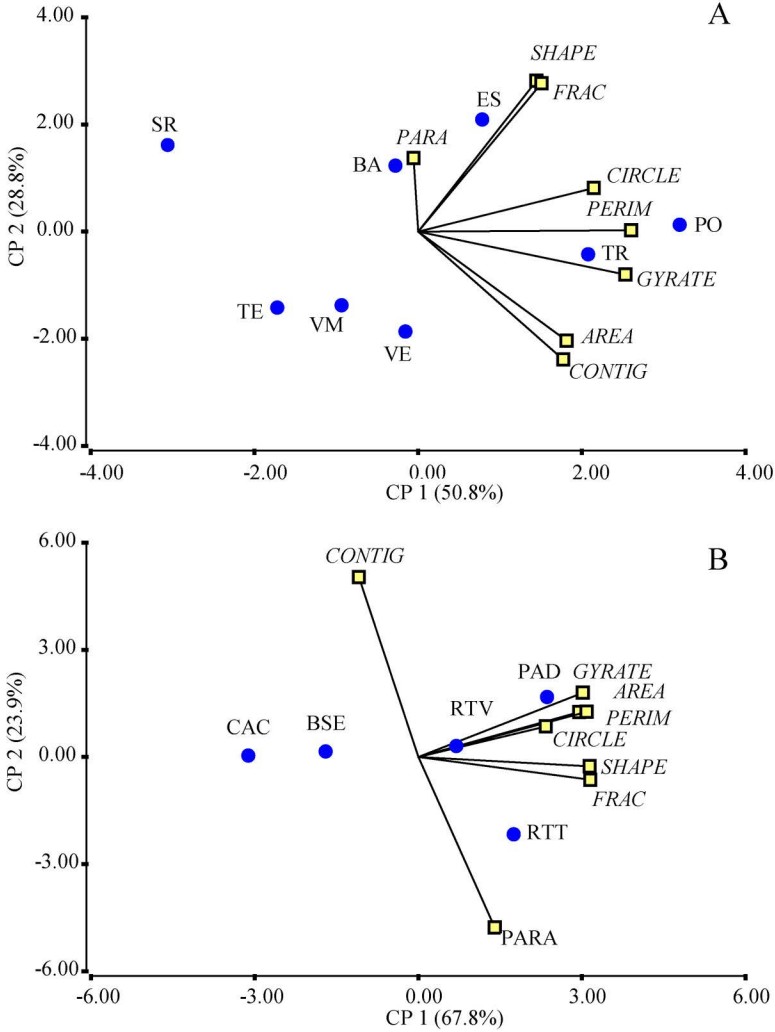

**Fig 3. Biplot generated from the principal component analysis (PCA) of landscape metrics.** For the mosaics (A) and vegetation covers (B) in landscapes with agroforestry and silvopastoral systems in the Colombian Amazon. Abbreviations: TR: El Triunfo; BA: Batalla 13; SR: Santa Rosa; TE: El Tesoro; PO: El Porvenir; VE: La Vega; VM: Villa Mery; ES: Esmeraldas; BSE: Secondary Forest; CAC: Agroforestry Cacao Cultivation; PAD: Pastures with Scattered Trees; RTT: Early Fallows; RTV: Old Fallows.

## Stored carbon in vegetation covers of the mosaics by system

A total of 78 temporary plots were established across 44 vegetation cover patches associated with agroforestry (23) and silvopastoral (21) landscape mosaics. Together, these two systems store a total of 12.16 Gg of carbon, contributing to the removal of 44.62 Gg of $CO_2$ emissions from the atmosphere across the 800 hectares sampled in this study. No significant differences were observed in average carbon storage between the systems ($F = 0.05$; $p = 0.8287$; $df = 77$; $\alpha = 0.05$); agroforestry systems (SAFc) contributed 15.27 Mg C ha$^{-1}$, while silvopastoral systems (SSP) contributed 15.13 Mg C ha$^{-1}$.

Carbon storage, expressed in megagrams per hectare (Mg ha$^{-1}$), in the different studied pools showed significant differences among mosaics only for herbaceous vegetation carbon (CAhe) ($F = 11.92$; $p = 0.0001$; $df = 7$) and litter carbon (CAho) ($F = 3.44$; $p = 0.0032$; $df = 7$), but not for aboveground and root biomass carbon (CAar). Regarding vegetation cover types, significant differences were observed in aboveground biomass carbon (CAar) ($F = 3.715$; $p = 0.0083$; $df = 4$), herbaceous

vegetation carbon (CAhe) (F = 3.87; p = 0.0066; df = 4), and litter carbon (CAho) (F = 3.30; p = 0.0152; df = 4). Secondary forests (BSE) showed the highest contribution, with 21.61 Mg C ha$^{-1}$ in aboveground biomass and a total carbon storage of 28.54 Mg ha$^{-1}$. In contrast, the cover type pastures with scattered trees (PAD) had the lowest accumulation of carbon in aboveground biomass and litter (8.02 Mg C ha$^{-1}$), but the highest in herbaceous vegetation (0.012 Mg C ha$^{-1}$) (Table 4).

Within the SAFc systems, the average carbon stored in the secondary forest (BSE) cover type (31.45 Mg ha$^{-1}$) differed significantly from all other cover types (F = 3.85; p = 0.0108; df = 4). No significant differences were found for carbon storage in the other pools evaluated within this system. In the SSP mosaics, herbaceous carbon storage differed significantly among cover types (F = 3.33; p = 0.0308; df = 3), while no significant differences were observed for litter or aboveground biomass carbon. In this system, the PAD and RTT cover types showed the highest average herbaceous carbon accumulation (0.01 Mg ha$^{-1}$), whereas the BSE cover type had the highest total carbon storage (18.34 Mg ha$^{-1}$).

## Relationship between stored carbon and landscape structure and configuration

In the partial least squares (PLS) regression analysis between metrics and carbon stored in the mosaics, factors 1 and 2 account for 61.1% of the variation (Fig 4). In the first factor, aboveground and litter carbon were positively associated with most of the SAFc mosaics, which were characterized by patches with smaller areas and higher connectivity. In contrast, herbaceous carbon was associated with the negative end of the axis, corresponding to mosaics with larger and more asymmetrical patches, typically found in SSP systems. The second factor grouped patch shape metrics and showed a negative association with both stored carbon and patch isolation metrics.

The PLS analysis between landscape metrics and carbon storage across vegetation cover types revealed that Factors 1 and 2 explained 61.1% of the variation (Fig 5). The first factor showed a positive relationship between carbon storage and patch area and connectivity metrics, and a negative relationship with patch shape and compactness metrics. The PAD cover type exemplifies this pattern, as it exhibited the highest values for AREA and CONTIG metrics. The second

**Table 4. Distribution of aboveground carbon – CAar, herbaceous carbon – CAhe, and litter carbon – CAho, stored in the mosaics and vegetation cover types of agroforestry and silvopastoral landscapes in the Colombian Amazon. Values sharing a common letter in their means are not significantly different (p > 0.05). Pairwise means comparison method: Fisher, alpha = 0.05.**

| | CAar (Mg ha$^{-1}$) | | | CAhe (Mg ha$^{-1}$) | | | CAho (Mg ha$^{-1}$) | | |
|---|---|---|---|---|---|---|---|---|---|
| | Mean | S.E. | | Mean | S.E. | | Mean | S.E. | |
| **Mosaics** | | | | | | | | | |
| SR | 21.08 | 2.46 | A | 0.0100 | 0.001 | A | 0.0100 | 0.001 | A |
| VE | 17.32 | 2.46 | A | 0.0036 | 0.001 | B | 0.0100 | 0.001 | A |
| PO | 16.35 | 2.46 | A | 0.0034 | 0.001 | BC | 0.0100 | 0.001 | AB |
| BA | 15.87 | 2.46 | A | 0.0022 | 0.001 | BC | 0.0049 | 0.001 | AB |
| VM | 15.57 | 2.76 | A | 0.0017 | 0.001 | BC | 0.0046 | 0.001 | AB |
| TR | 13.85 | 2.46 | A | 0.0016 | 0.001 | BC | 0.0046 | 0.001 | AB |
| ES | 12.00 | 2.46 | A | 0.0011 | 0.001 | BC | 0.0030 | 0.001 | BC |
| TE | 11.97 | 2.46 | A | 0.0006 | 0.001 | C | 0.0019 | 0.001 | C |
| **Vegetation covers** | | | | | | | | | |
| BSE | 21.61 | 2.64 | A | 0.0009 | 0.002 | B | 0.0100 | 0.001 | A |
| CAC | 17.35 | 2.15 | AB | 0.0020 | 0.001 | AB | 0.0036 | 0.001 | AB |
| PAD | 8.02 | 3.04 | C | 0.0100 | 0.002 | A | 0.0015 | 0.001 | B |
| RTT | 13.87 | 1.26 | BC | 0.0048 | 0.001 | A | 0.0048 | 0.000 | A |
| RTV | 17.31 | 1.81 | AB | 0.0009 | 0.001 | B | 0.0100 | 0.001 | A |

BA: Batalla 13; TE: El Tesoro; TR: El Triunfo; SR: Santa Rosa; ES: Esmeraldas; PO: El Porvenir; VE: La Vega; VM: Villa Mery; BSE: Secondary forest; CAC: Agroforestry cacao cultivation; PAD: Pastures with scattered trees; RTT: Early Fallows; RTV: Old Fallows.

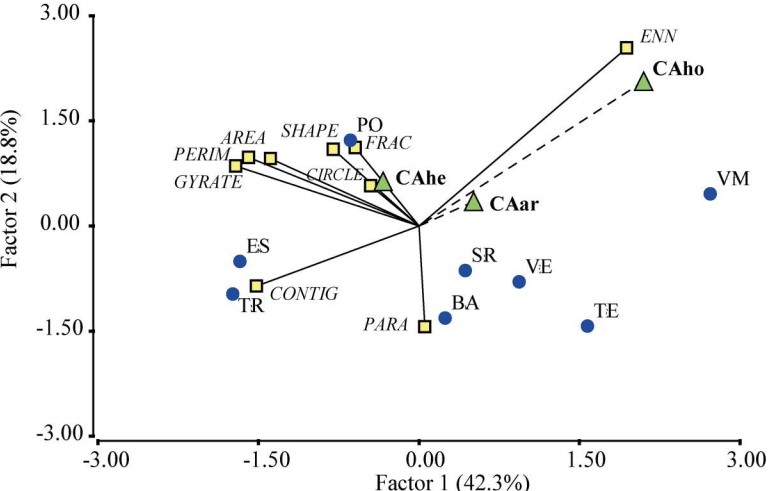

**Fig 4. Triplot obtained through partial least squares (PLS) regression between carbon, metrics, and landscape mosaics in agroforestry and silvopastoral systems in the Colombian Amazon.** TR: El Triunfo; BA: Batalla 13; SR: Santa Rosa; TE: El Tesoro; PO: El Porvenir; VE: La Vega; VM: Villa Mery; ES: Esmeraldas.

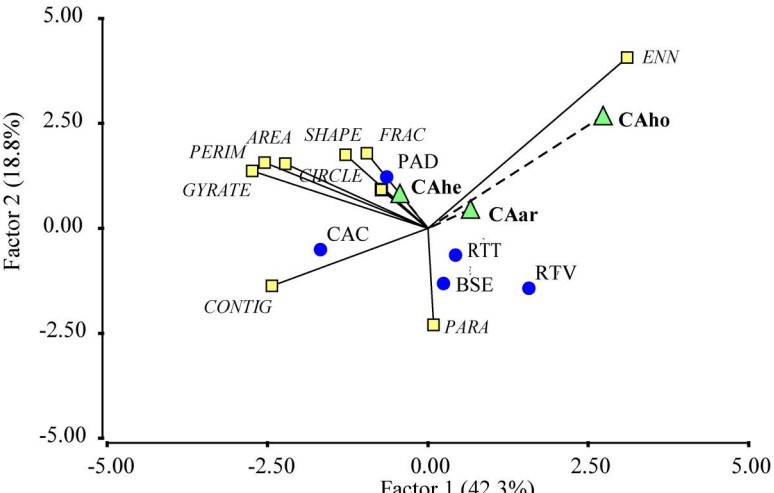

**Fig 5. Triplot obtained through partial least squares (PLS) regression between carbon, metrics, and vegetation covers in agroforestry and silvopastoral systems in the Colombian Amazon.** BSE: Secondary forest; CAC: Agroforestry cocoa crop; PAD: Pastures with dispersed trees; RTT: Early fallows; RTV: Old fallows.

factor revealed a negative association between patch shape metrics and carbon stored across all components, as well as with isolation metrics between patches.

## Discussion

### Analysis of landscape structure and configuration in cacao agroforestry and silvopastoral systems

In the Colombian Amazon region, this study represents the first comprehensive analysis to evaluate how the spatial structure and configuration of the landscape influence the ecosystem service of carbon storage in cacao agroforestry systems

(SAFc) and silvopastoral systems (SSP). The findings provide critical insights to guide land use management and planning at the landscape scale. Key attributes such as structural heterogeneity, patch size, shape, and degree of aggregation were found to influence carbon sequestration, biomass production, and the mitigation of greenhouse gas emissions.

The overall landscape structure analysis by mosaic revealed subtle differences between the systems in terms of patch shape, size, and aggregation. SAFc mosaics were characterized by smaller average patch areas, more complex and elongated shapes with irregular edges, but high spatial aggregation. In contrast, SSP mosaics exhibited greater variability in structural metrics, with patches that were more heterogeneous in size and shape. These differences suggest a direct influence of land-use management, where SAFc systems maintain a more cohesive configuration linked to diversified agroforestry practices, while SSP systems reflect processes of fragmentation, secondary regeneration, and extensive livestock grazing [6,61].

The relationship between types of vegetation cover and landscape metrics revealed variations in the shape, size, and spatial arrangement of patches. Secondary vegetation fallows and pastures with scattered trees exhibited the largest extents, although they were associated with irregular shapes and low aggregation. In contrast, covers with higher tree density, such as cacao agroforestry plots and secondary forests, were characterized by smaller patches with more regular contours but greater isolation. These findings indicate that the spatial structure of the landscape varies according to vegetation cover type. At the same time, the configuration suggests fragmentation associated with extensive land use and limited spatial planning in productive landscapes, where patches often remain in intermediate stages of ecological succession [62]. Furthermore, the results point to greater structural integrity at the patch level, which, according to Tiang et al. [63], may promote the local conservation of ecological functions such as carbon sequestration and biodiversity. However, this potential is constrained by the limited connectivity between patches [26].

## Carbon stored in vegetation covers across landscape mosaics by system

According to Salete-Capellesso et al. [64], carbon dynamics in tropical forests are influenced by a range of biotic and abiotic factors acting across spatial and temporal scales. Similarly, Suárez et al. [65] suggest that, in the Amazon, biomass production and carbon stocks vary across land uses due to differences in the composition and structure of agroforestry and silvopastoral systems. Although this study did not find statistically significant differences in carbon storage between systems, SAFc landscapes stored slightly more carbon. Mosaics in this system included vegetation patches with greater tree density.

Agudelo-Hz et al. [2] and Mendes Pereira et al. [66] suggest that higher tree density and diversity, along with sustainable management practices, enhance tree biomass growth and accumulation [20]. In this study, cacao agroforestry systems demonstrated functional integration between woody species and cacao crops. This finding aligns with Clemente-Arenas [67] and Surco-Huacachi & Garate-Quispe [68], who argue that such integration enhances the provision of regulatory ecosystem services such as carbon sequestration and storage [40]. Solarte et al. [69] suggest that the presence of multiple permanent tree strata in these systems increases both above- and below-ground biomass while maintaining ecological processes such as microclimate regulation, nutrient cycling, and the provision of structurally complex habitats [7]. According to Bethwell et al. [70], the multifunctionality of SAFc positions them as a viable strategy for biodiversity conservation at the landscape scale, climate change mitigation, and ecological restoration in transformed Amazonian landscapes [71].

The amount of carbon stored in SAFc estimated in this study is comparable to the results obtained by Carvajal-Agudelo & Andrade [31] in Casanare (Colombia – CO), who reported 16.0 Mg C ha$^{-1}$, and by Leiva-Rojas and Ramírez-Pisco [32], who reported between 14.13 and 14.33 Mg C ha$^{-1}$ for SAF systems aged 10–30 years in Antioquia (CO). Moreover, the values reported here are three times higher than those found by Mena and Andrade [30] in SAFc systems in Chocó (CO). However, they remain lower than the 59.8 Mg C ha$^{-1}$ of aboveground biomass recorded by Hernández Núñez et al. [18] in SAFc systems in Meta (CO). These differences may result from the complex interaction of biophysical, edaphic, and management factors that modulate primary productivity and the carbon storage capacity of agroforestry systems [72].

According to Ordoñez & Rangel-Ch [73], the structure and floristic composition of vegetation cover in SAFc systems are crucial determinants of carbon fixation and storage. In the Colombian Amazon, agroforestry systems integrate a high diversity of native flora, including timber species, cacao shrubs, and leguminous trees [74], contributing to higher structural complexity [75]. This functional and structural complexity increases vertical and horizontal heterogeneity [76], with coexisting tree, shrub, and herbaceous strata, diversification of ecological niches, and forest species exhibiting high resource-use efficiency [77]. Studies reported by Gonzalez & Duff [78] and Quinto-Mosquera [79] also suggest these systems possess greater leaf area and increased net primary productivity, while also generating more stable microclimatic conditions by regulating internal temperature, humidity, and light levels [80,81]. In this context, although the carbon stocks in SAFc remain lower than those reported by Díaz-C et al. [3] in disturbed Amazonian forests (142.09 Mg C ha$^{-1}$), the implementation of agroforestry systems nonetheless contributes significantly to carbon sequestration and strengthens resilience to historical disturbances in the region [82].

The carbon stocks observed in SSP systems in this study exceed those reported in other research conducted in Caquetá (CO): Villegas et al. [39] reported 8.69 Mg C ha$^{-1}$, Rojas-Vargas et al. [35] found 2.59 Mg C ha$^{-1}$, and Pardo-Rozo et al. [38] reported 1.40 Mg C ha$^{-1}$. However, the values observed here are lower than those found in SSP systems associated with forest regeneration areas in Colombia's Orinoquía region [18] and the Colombian Amazon [35]. According to Salete-Capellesso et al. [64], the regeneration of secondary vegetation patches increases carbon sequestration potential due to the progressive rise in structural complexity and the longer residence time of carbon in woody biomass. In this regard, the carbon storage capacity of silvopastoral systems varies with the degree of anthropogenic disturbance (63) and the implementation of management practices [83]. According to Rodríguez-León et al. [84], the diversity of forest species in regenerating areas and the ecological maturity of the mosaic [85] are key factors influencing biomass production and carbon storage in livestock-dominated landscapes [86–88].

## Relationship between carbon storage and landscape structure and configuration

Distinct patterns were observed in the relationship between landscape metrics, mosaic structures, and carbon storage across the components analyzed. Carbon stored in aboveground biomass and leaf litter was not affected by variations in patch size or shape. However, it showed a positive association with structural connectivity in mosaics composed of agroforestry cacao systems (SAFc). Specifically, carbon storage increased in mosaics with higher contiguity among patches of the same land cover class and decreased in those with greater contrast. In line with previous findings [89], this result suggests that enhanced spatial continuity among vegetation patches promotes ecological processes such as energy flow, moisture retention, and nutrient cycling efficiency [90]. These processes contribute to increased plant productivity and, consequently, higher carbon fixation and storage in SAFc systems [91–93]. Additionally, carbon stored in aboveground biomass and leaf litter decreased as mosaic fragmentation intensified. According to Rosan et al. [94], highly fragmented landscapes display discontinuous spatial structures and more pronounced edge effects, which disrupt microclimatic conditions, reduce ecological flows, and undermine the capacity to sustain large, long-lived woody species—key contributors to carbon accumulation [95]. A mosaic with higher contrast between a given patch and its surrounding patches indicates a more fragmented landscape [96–98]. These findings are consistent with those of Lamy et al. [48], who reported that habitat fragmentation reduces forest biomass due to higher tree mortality at patch edges and the dominance of pioneer or disturbance-adapted species, which typically have shorter life cycles and lower carbon fixation capacity.

Additionally, the results revealed a positive relationship between carbon stored in the herbaceous layer and patch area in mosaics dominated by silvopastoral systems (SSP), while a negative relationship was observed with shape and aggregation metrics. Specifically, herbaceous carbon tended to increase in larger, more irregularly shaped patches, often found in landscapes with low connectivity. According to Serna et al. [93], this pattern may be associated with extensive land use and canopy opening in SSPs, which favor the dominance of grasses [99,100] and pioneer species characterized by high growth rates and rapid biomass accumulation in the lower strata [86,101]. However, Chisholm & Gray [102] suggest that

this accumulation is more indicative of temporary cover dynamics than of long-term carbon stabilization processes, underscoring the importance of evaluating carbon quality and permanence in fragmented landscape contexts [71,103].

The patches within agroforestry cacao system (SAFc) mosaics showed greater connectivity with forested and fallow areas, which tended to be smaller in size and more circular in shape. This structural condition contributed to increased forest density, biomass production, and carbon storage by enhancing habitat heterogeneity [104]. In contrast, SSP mosaics exposed to more intensive land use—such as extensive cattle grazing—exhibited greater fragmentation, isolation, and a loss of forest cover, ultimately reducing the abundance of woody vegetation. Agroforestry systems can be strategically managed to support both plant biodiversity conservation and sustainable agricultural production [14]. According to Schloss et al. [105], landscapes under productive management can serve as buffers between intensively used areas and biologically significant regions.

In the agroforestry cacao (SAFc) and silvopastoral systems (SSP) of the Colombian Amazon, a positive relationship was observed between landscape metrics and the ecosystem service of carbon regulation. Mosaics with greater structural heterogeneity, functional connectivity, and the presence of natural vegetation covers stored more carbon compared to those composed of isolated patches with complex morphology. These results support the hypothesis that landscape configuration and composition affect the provision of ecosystem services, aligning with findings by Lamy et al. [48] for forest–agriculture transitional landscapes.

The results of this study confirm that the spatial configuration of the landscape influences carbon sequestration and storage [105,106], with lower values found in mosaics with reduced patch connectivity and higher structural fragmentation [107,108]. This underscores the vulnerability of these systems to land-use and land-cover changes [85,109,110]. From a conservation and sustainable management perspective, these findings highlight the ecological and productive value of maintaining and enhancing both structural and functional connectivity among natural vegetation patches, as well as integrating forested covers within agroforestry and silvopastoral matrices characterized by high structural diversity.

Therefore, it is necessary to advance research at multiple spatial and temporal scales to evaluate resource use patterns and interspecific dynamics under changing landscape configuration scenarios. It is also recommended that productive landscape management in the Colombian Amazon adopt an integrated landscape-scale approach. This approach should include actions aimed at increasing structural heterogeneity within mosaics, designing larger, more circular patches with greater spatial contiguity, and implementing sustainable management practices such as the diversification of native woody species, enhancing vertical and horizontal vegetation structure, and restoring ecological corridors. Such management not only enhances the provision of regulating ecosystem services—such as carbon sequestration and storage—but also offers economic alternatives to cacao and livestock producers, through access to conservation incentives such as Payments for Environmental Services (PES) and carbon credit certification.

## Conclusions

The analysis of production system mosaics revealed that the carbon sequestration ecosystem service depends on the structure and spatial configuration of landscapes in agroforestry systems with cacao and silvopastoral systems. Mosaics with a greater number and diversity of patches exhibited higher biomass production and carbon storage. Additionally, patches with symmetrical, less irregular shapes and stronger connectivity between forested and agroforestry areas facilitate carbon exchange among landscape components.

This study highlights the need for integrated management of rural landscapes in the Colombian Amazon, emphasizing elements related to landscape structure, composition, and spatial configuration. Effective planning should consider the arrangement and size of vegetation patches, diversity and type of vegetation cover, and distribution and connectivity among patches. This approach will not only enhance carbon capture and storage but also promote the environmental and economic sustainability of these systems, contributing to the conservation of ecosystem services and mitigation of climate change effects.

## Supporting information

**Inclusivity in global research questionnaire.  Inclusivity in global research questionnaire.**
(PDF)

**S1–S4 Tables.  Supplementary material.**
(DOCX)

## Acknowledgments

The authors thank the INBIANAM Center for providing equipment for fieldwork. To the Vicerrectoría de Investigación e Innovación of the Universidad de la Amazonia – Colombia, for its support in partially funding the doctoral program. We also express our gratitude to the Comité Departamental de Ganaderos del Caquetá and the Asociación Departamental de Cultivadores de Cacao y Especies Maderables del Caquetá (ACAMAFRUT) for facilitating access to the properties. Special thanks to Alejandro Navarro for creating the thematic land cover map and to the BySE Research Group for their support during field data collection.

## Author contributions

**Conceptualization:** Jenniffer Díaz-Cháux.

**Data curation:** Jenniffer Díaz-Cháux, Alexander Velasquez-Valencia.

**Funding acquisition:** Jenniffer Díaz-Cháux, Alexander Velasquez-Valencia.

**Investigation:** Jenniffer Díaz-Cháux.

**Resources:** Jenniffer Díaz-Cháux, Alexander Velasquez-Valencia.

**Supervision:** Alexander Velasquez-Valencia.

**Validation:** Jenniffer Díaz-Cháux, Alexander Velasquez-Valencia, Fernando Casanoves.

**Writing – original draft:** Jenniffer Díaz-Cháux.

**Writing – review & editing:** Jenniffer Díaz-Cháux, Alexander Velasquez-Valencia, Fernando Casanoves.

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
