## [Decision Letter · Decision Letter 0]

PONE-D-24-52526Influence of Landscape Structure on Carbon Storage in Agroforestry Systems with Cacao and Silvopastoral Systems in the Colombian AmazonPLOS ONE

Dear Dr. Diaz Chaux,

Thank you for submitting your manuscript to PLOS ONE. After careful consideration, we feel that it has merit but does not fully meet PLOS ONE’s publication criteria as it currently stands. Therefore, we invite you to submit a revised version of the manuscript that addresses the points raised during the review process.

The authors conducted a systematic review of the effects of Conservation Reserve Program in USA. They systematically searched for studies dealing with Soil/vegetation/wildlife/social and other aspects that were affected by CRP. They also conducted spatial analyses to show which states participating in the program have conducted studies to assess the effects of CRP. The study was done well and summaries have been provided on how many studies were conducted on different habitat/organismal factors affected by CRP. However, the authors did not provide good syntheses in the Discussion. For example, there is good summaries on how many studies covered soil improvement and from which State(s), but there are limited or no syntheses extracted from these studies as discussion points. The same is true of vegetation, wildlife, social studies etc. I feel the paper would improve a lot more if such information is summarized and synthesized from the substantial numbers of papers to enrich the discussion. I suggest major review and i have appended a marked up version of the manuscript where I propose these changes need to be incorporated.

We look forward to receiving your revised manuscript.

Kind regards,

Marcela Pagano, Ph.D, M.D.

Academic Editor

PLOS ONE

Journal Requirements:

3. Thank you for stating the following financial disclosure: [Beca de excelencia doctoral Bicentenario otorgada por Minciencias Colombia (código BPIN 2021000100036)].

Please state what role the funders took in the study. If the funders had no role, please state: "The funders had no role in study design, data collection and analysis, decision to publish, or preparation of the manuscript.

4.We note that your Data Availability Statement is currently as follows: [Todos los datos relevantes están dentro del manuscrito y de sus archivos de información de apoyo]

Additional Editor Comments:

Dear authors, please, revise your manuscript according to reviewer suggestions.

Reviewers' comments:

Reviewer's Responses to Questions

**Comments to the Author**

1. Is the manuscript technically sound, and do the data support the conclusions?

Reviewer #1: Yes

Reviewer #2: Yes

Reviewer #3: Partly

2. Has the statistical analysis been performed appropriately and rigorously? 

Reviewer #1: Yes

Reviewer #2: Yes

Reviewer #3: I Don't Know

3. Have the authors made all data underlying the findings in their manuscript fully available?

Reviewer #1: Yes

Reviewer #2: Yes

Reviewer #3: No

4. Is the manuscript presented in an intelligible fashion and written in standard English?

Reviewer #1: Yes

Reviewer #2: Yes

Reviewer #3: Yes

5. Review Comments to the Author

Reviewer #1: In the manuscript, the authors did not include much data and objective in the abstract.

There is also a need to incorporate the objective in the introduction section.

Please write the aim and applicability of the study.

There is a need to include study site photographs for authentication and to clarify the hypothesis of this study.

Please add many more recent references related to the study.

I strongly suggest that the author edit and rewrite the sentences for language editing, which is the most useful criterion of any scholarly research article.

Discussion of the manuscript needs to elaborate by including some recent literature.

Reviewer #2: I reviewed the manuscript titled "Influence of Landscape Structure on Carbon Storage in Agroforestry Systems with Cacao and Silvopastoral Systems in the Colombian Amazon" for consideration in PLOS ONE. This study addresses a critical issue in the conservation of Amazonian ecosystems by evaluating the influence of landscape structure and configuration on carbon storage in cacao agroforestry systems (SAFc) and silvopastoral systems (SSP). The study provides valuable insights into the role of agroforestry mosaics in carbon regulation, highlighting their potential for climate change mitigation. The research suggests strategies to enhance connectivity and increase carbon sequestration capacity in productive landscapes, which has significant implications for land-use planning. However, I do have some suggestions and areas for improvement for the manuscript:

• Biomass and carbon storage may vary depending on the floristic composition within SAFc and SSP, suggesting the need for further studies on the contribution of key species.

• Differences in management practices within agroforestry systems could affect carbon dynamics, which could be explored in greater depth.

• The statistical analysis and results section presents valuable insights, but there are several areas where clarity, coherence, and interpretation could be improved.

• The results section presents p-values and F-statistics but lacks biological or ecological interpretation of the findings. I suggest adding means and SD of each variable to give a better perspective of the values and differences and explain the ecological significance of significant and non-significant results.

• Add a brief explanation of why each statistical method was chosen and how it aligns with the study objectives.

• The manuscript does not mention whether key assumptions for ANOVA (normality, homogeneity of variance) and regression analyses (linearity, multicollinearity) were checked.

• Some tables and figures lack clear captions or explanations, making it difficult for readers to interpret the data. Please improve the clearly label significant results with asterisks or superscripts in tables. Ensure figure legends describe all abbreviations and variables. For example, table 3 presents mean carbon storage values across different vegetation covers. Add a legend indicating that values with the same letter are not significantly different (Tukey’s HSD test, p > 0.05).

• The results do not mention potential confounding factors that could influence carbon storage variability (e.g., soil type, elevation, past management). Please address potential confounders and, if applicable, describe how they were controlled.

• The PCA and PLS regression results are presented with numerical outputs do not have a clear biological interpretation. Please explain what the principal components represent and how the regression findings contribute to the study’s conclusions.

• The discussion section provides a general overview of the findings, but it would benefit from a more in-depth interpretation of the observed patterns.

• I suggest explicitly addressing how different landscape metrics influence carbon sequestration and why certain patch configurations lead to higher/lower storage.

• The discussion includes citations to previous studies but lacks a critical comparison that highlights the novelty of the current research. Clearly articulate how the results confirm, contradict, or expand on prior studies.

• Although different landscape mosaics are analyzed, microclimatic and edaphic factors that may influence carbon capture are not explored in detail.

• The discussion would benefit from explicitly acknowledging limitations and suggesting future research directions.

• The discussion highlights the importance of landscape connectivity but does not explicitly state how land-use policies should integrate these findings. Please expand the implications by providing concrete recommendations for conservation planning

Reviewer #3: #Revisor

The manuscript deals with an interesting subject involving landscape metrics and measurements of biomass and carbon in vegetation in the field. The authors indicate that this type of study is a pioneer in the region. However, I found the article difficult to understand for a general audience, and it had a weak connection between the objectives, the methodology, and the results. I have tried to highlight some of the critical points below. A conceptual issue that seems important to me is the classification of silvopastoral systems, which involve trees, pastures, and animals. The system closest to the silvopastoral system (PAD) is poorly represented. So, it does not seem easy to relate landscape metrics to the carbon stored in production systems (SAF; SPP). In addition, some results are conflicting, such as the carbon in the aerial biomass of grasses being higher than in secondary tropical forests; estimation of carbon stored in the biomass of trees, herbaceous, and litter in LAG=lagoons; use of biomass equations for fruit species, but they are not highlighted in any production system. In addition, the manuscript needs to reorganize the text and its writing. I could not recommend the acceptance of the manuscript in its present form.

Material and methods

The separation of the M&M item could be redone. Line 148 onwards should not be part of the study area.

L148-149. Could it include the main species that make up the SSP? Does SAF with cocoa present high or low plant diversity?

L170-173. What would be the area of each of the 78 plots? How were the plots within each mosaic systematized?

L99-211. This topic is confusing. How was the carbon stored in each plot converted to SAF, SPP, and mosaic model scale? Did the authors use the same equation to convert aerial biomass into root biomass for all vegetation types?

Why were landscape metrics classified as descriptive to use in PCA if they are quantitative in regression? Moreover, why do linear regression and PLS? Are the authors using the same metrics as Table 1 for all cases (PCA, LR, SLS)? How did the authors deal with the different measurement scales to do the PCA? In general, PCA can be influenced by high and low values.

Results

A more coherent separation of the results into topics would facilitate understanding.

It is unclear what type of ecosystem fits the silvopastoral systems (SPP) classification. The only one that could fit is the PAD model, represented only by the ES Mosaic. So, I asked how it was possible to statistically test the effects of the SAF and SPP models on the stored carbon. Also, how did the authors calculate the carbon stored in trees in LAG, RIO, PEN, PPH, PPL, RTT, and RTV?

Table 3. The table is cut off, and I cannot see all the information. Furthermore, what is the unit of values? Is the size of the area in ha, m2, km2? Why isn't there SD in some of the information? Why are the letters that indicate statistical results only in CAC and PPL?

L230-245. The authors indicate the different landscape metrics in the PCA, but it would be easier to relate them to Table 1. For example, which metrics are related to "complexity in patch shape, while the negative end associates with mosaics featuring larger, more aggregated patches" "mosaics with irregular, aggregated patches, while the negative end is characterized by mosaics with larger patches that have smoother, less irregular contours".

L255-264. This paragraph should be in the M&M. In addition, the authors could go into more detail about this type of information:

"The cover type with the highest number of plots was early fallow (RTT), followed by old fallow (RTV) and agroforestry cacao cultivation (CAC). A total of 2,846 individual plants were surveyed across both systems: 1,505 trees in the SAFc mosaics, including 330 cacao shrubs, 105 rubber trees (H. brasiliensis), 29 palms, thirteen fruit trees, and 1,028 saplings."

L266-276. Given the diversity of ecosystems and the very different number of replicates, comparing the SAF and SPP systems seems difficult. In addition, it seems strange to me that the biomass of grasses is greater than forest (BSE). The carbon stored in the soil could be higher in grass than in secondary forest, but I had not yet seen biomass.

Fig.3. The figure does not show ANOVA, but rather the average carbon values in biomass. A period is used and not a comma to separate the numbers in English.

L289. How did the authors arrive at this value of stored carbon?

L305-303. Is carbon stored in biomass equal to total carbon? Usually, grasses have a lot of root biomass.

Discussion

L419. This connectivity metric is unclear. Where is this result?

L 428-432. This relationship was weak many times for some metrics. Could you explain which metrics might be most important for estimating carbon relative to Table 1? Although the linear regression was significant, the data distribution in Fig 4 does not appear linear.

6. PLOS authors have the option to publish the peer review history of their article (what does this mean? ). If published, this will include your full peer review and any attached files.

**Do you want your identity to be public for this peer review?** For information about this choice, including consent withdrawal, please see our Privacy Policy .

Reviewer #1: No

Reviewer #2: No

Reviewer #3: No

---

## [Author Response · Author response to Decision Letter 1]

1 May 2025

Florencia, Colombia, April 14, 2025

Dra.

Marcela Pagano, Ph.D, M.D.

Academic Editor

PLOS ONE

Greetings,

Respectfully, the authors of the manuscript titled "Influence of Landscape Structure on Carbon Storage in Agroforestry Systems with Cacao and Silvopastoral Systems in the Colombian Amazon" hereby submit the "Response to Reviewers" letter for your consideration. In this document, we provide detailed responses to each of the questions, suggestions, and comments raised during the review process. o facilitate the understanding of this document, we would like to inform that the responses to each question are provided at the end of each section, highlighted in bold text. We trust that we have been thorough and detailed in addressing the queries.

Reviewer #1

In the manuscript, the authors did not include much data and objective in the abstract. Response: The abstract has been revised to explicitly include the overarching objective of the study. Furthermore, additional data have been incorporated into the manuscript to enhance the robustness and clarity of the presented results.

There is also a need to incorporate the objective in the introduction section. Response: we have included the overall objective of the study at the end of the Introduction section to provide greater clarity regarding the scope and intent of the research.

Please write the aim and applicability of the study. Response: The overall objective of the research has now been explicitly stated in both the Abstract and the Introduction sections to enhance clarity and align with the manuscript’s scientific narrative.

There is a need to include study site photographs for authentication and to clarify the hypothesis of this study. Response: Figure 2 has been incorporated to include representative photographs of the different types of vegetation cover identified within the eight landscape mosaics characterized by cacao agroforestry systems (AFS) and silvopastoral systems (SPS) studied. These images aim to enhance the visual understanding of the structural heterogeneity and composition of the land cover types across the sampled landscapes.

Please add many more recent references related to the study. Response: We conducted a more in-depth literature review to identify studies as closely related as possible to the objectives of our research. As a result, we have included additional references and incorporated further biological explanations to support the interpretation of our findings. It is important to clarify that, for our study area—the Colombian Amazon—this work represents a pioneering effort in analyzing, at a landscape scale, the causal relationships between landscape structure and configuration and an ecosystem service of high relevance: carbon sequestration. This is particularly significant in the context of climate change mitigation initiatives.

I strongly suggest that the author edit and rewrite the sentences for language editing, which is the most useful criterion of any scholarly research article. Response: A thorough revision of the manuscript was conducted, with particular attention to improving the overall writing style. This effort aimed to ensure a high level of coherence and clarity in the presentation of ideas throughout the text.

Discussion of the manuscript needs to elaborate by including some recent literature. Response: In collaboration among the three authors, the presentation of the results has been restructured. This revision allowed for a more comprehensive analysis and an expanded discussion, now supported by a broader range of relevant literature and recent studies.

Reviewer #2

I reviewed the manuscript titled "Influence of Landscape Structure on Carbon Storage in Agroforestry Systems with Cacao and Silvopastoral Systems in the Colombian Amazon" for consideration in PLOS ONE. This study addresses a critical issue in the conservation of Amazonian ecosystems by evaluating the influence of landscape structure and configuration on carbon storage in cacao agroforestry systems (SAFc) and silvopastoral systems (SSP). The study provides valuable insights into the role of agroforestry mosaics in carbon regulation, highlighting their potential for climate change mitigation. The research suggests strategies to enhance connectivity and increase carbon sequestration capacity in productive landscapes, which has significant implications for land-use planning. However, I do have some suggestions and areas for improvement for the manuscript:

• Biomass and carbon storage may vary depending on the floristic composition within SAFc and SSP, suggesting the need for further studies on the contribution of key species. Response: As noted in the methodology section, we included a general description of the composition and structure of the sampled cacao-based agroforestry systems and silvopastoral systems in the Colombian Amazon. This characterization aimed to provide essential context for interpreting the ecological dynamics and functional patterns observed across the studied land-use systems.

• Differences in management practices within agroforestry systems could affect carbon dynamics, which could be explored in greater depth. Response: In the description of the systems within the study area, we included a detailed account of the management characteristics specific to each system. This information was incorporated to provide a comprehensive understanding of the context in which each system operates, thereby supporting the interpretation of the results within their respective socio-ecological frameworks.

• The statistical analysis and results section presents valuable insights, but there are several areas where clarity, coherence, and interpretation could be improved. The results section presents p-values and F-statistics but lacks biological or ecological interpretation of the findings. I suggest adding means and SD of each variable to give a better perspective of the values and differences and explain the ecological significance of significant and non-significant results. Response: Both the description of the statistical analysis section and the presentation of the results have been revised to include the means, standard deviations, and standard errors for each analyzed variable. Additionally, the supplementary materials file has been consolidated to present the results of the conducted analyses, as well as the matrix of the field-measured dendrometric variables.

• Add a brief explanation of why each statistical method was chosen and how it aligns with the study objectives. Response: The necessary adjustments have been made to the statistical analysis section of the manuscript in accordance with the feedback provided.

• The manuscript does not mention whether key assumptions for ANOVA (normality, homogeneity of variance) and regression analyses (linearity, multicollinearity) were checked. Response: The statistical analysis section of the manuscript has been reorganized and adjusted in accordance with the reviewers' suggestions. We believe these modifications enhance the clarity and rigor of the analyses presented.

• Some tables and figures lack clear captions or explanations, making it difficult for readers to interpret the data. Please improve the clearly label significant results with asterisks or superscripts in tables. Ensure figure legends describe all abbreviations and variables. For example, table 3 presents mean carbon storage values across different vegetation covers. Add a legend indicating that values with the same letter are not significantly different (Tukey’s HSD test, p > 0.05). Response: We have made the necessary adjustments to the statistical analyses in the corresponding section. Additionally, we refined the presentation of the results. Several figures that were previously unclear in conveying the results have been modified, and their descriptions have been elaborated in the figure legends to enhance the clarity and interpretation of the data presented.

• The results do not mention potential confounding factors that could influence carbon storage variability (e.g., soil type, elevation, past management). Please address potential confounders and, if applicable, describe how they were controlled. Response: In the expanded and revised discussion of the results, we have addressed potential factors that may influence carbon storage. This revision includes a comprehensive analysis of the underlying variables, incorporating both biotic and abiotic elements that could significantly affect carbon sequestration in the studied system.

• The PCA and PLS regression results are presented with numerical outputs do not have a clear biological interpretation. Please explain what the principal components represent and how the regression findings contribute to the study’s conclusions. Response: We have taken into account the potential confusion arising from the presentation of both PCA and PLS regression analyses. Consequently, we have prioritized the inclusion of the PLS triplot, as it effectively combines the features of Principal Component Analysis (PCA) and Multiple Regression, offering a more comprehensive representation of the data. We believe this approach enhances clarity and provides a more integrated view of the results.

• The discussion section provides a general overview of the findings, but it would benefit from a more in-depth interpretation of the observed patterns. Response: In response to the valuable and insightful recommendations provided by the reviewers, a comprehensive literature search was conducted, and the analysis of the results presented in the manuscript has been significantly strengthened. We have carefully incorporated the latest relevant studies to ensure a robust and thorough discussion of the findings. The revisions aim to enhance the clarity and depth of the analysis, aligning it with current advancements in the field.

• I suggest explicitly addressing how different landscape metrics influence carbon sequestration and why certain patch configurations lead to higher/lower storage. Response: In line with the central objective of this study, we clearly addressed the relationship between carbon storage components, landscape mosaics, and the vegetative covers classified within the agroforestry systems involving cacao and silvopastoral systems. We also examined the metrics that define the structure and spatial configuration of the studied landscapes.

• The discussion includes citations to previous studies but lacks a critical comparison that highlights the novelty of the current research. Clearly articulate how the results confirm, contradict, or expand on prior studies. Response: The authors believe that this recommendation has been adequately addressed. A coherent explanation of the results was provided, with a thorough comparison to previous studies related to both the subject and the object of investigation. Furthermore, these results were contrasted with studies conducted in the specific region of interest, ensuring a comprehensive context for the findings.

• Although different landscape mosaics are analyzed, microclimatic and edaphic factors that may influence carbon capture are not explored in detail. Response: In the adjusted results discussion, we included an analysis of the environmental and climatic variables of the region that are influencing, and may also be affected by, changes in the spatial structure and configuration of the mosaics with SAF and SSP.

• The discussion would benefit from explicitly acknowledging limitations and suggesting future research directions. Response: In the discussion, we have included several general recommendations regarding landscape management, emphasizing the importance of maintaining structural connectivity and addressing fragmentation issues. These measures are aimed at enhancing the provision and/or maintaining the regulation of ecosystem services, such as carbon storage, which is critical for sustaining ecological functions and mitigating climate change.

• The discussion highlights the importance of landscape connectivity but does not explicitly state how land-use policies should integrate these findings. Please expand the implications by providing concrete recommendations for conservation planning. Response: The manuscript includes considerations regarding the management and governance of rural landscapes, particularly focusing on the spatial structure and configuration of landscapes within agroforestry systems involving cacao and silvopastoral systems in the Amazon region. These aspects are critical for understanding how the integration of such systems can influence both ecological dynamics and sustainable land use practices in this unique and vital biome.

Reviewer #3:

The manuscript deals with an interesting subject involving landscape metrics and measurements of biomass and carbon in vegetation in the field. The authors indicate that this type of study is a pioneer in the region. However, I found the article difficult to understand for a general audience, and it had a weak connection between the objectives, the methodology, and the results. I have tried to highlight some of the critical points below. A conceptual issue that seems important to me is the classification of silvopastoral systems, which involve trees, pastures, and animals. The system closest to the silvopastoral system (PAD) is poorly represented. So, it does not seem easy to relate landscape metrics to the carbon stored in production systems (SAF; SPP). In addition, some results are conflicting, such as the carbon in the aerial biomass of grasses being higher than in secondary tropical forests; estimation of carbon stored in the biomass of trees, herbaceous, and litter in LAG=lagoons; use of biomass equations for fruit species, but they are not highlighted in any production system. In addition, the manuscript needs to reorganize the text and its writing. I could not recommend the acceptance of the manuscript in its present form. Response: We sincerely appreciate your thoughtful comment, which has greatly contributed to the substantial improvements in both the presentation of our results and the analysis provided. We believe that the revisions made address your concerns effectively and that the manuscript now meets the scientific rigor and the submission guidelines outlined for authors. We are confident that these adjustments make the paper suitable for publication in the journal.

Material and methods

The separation of the M&M item could be redone. Line 148 onwards should not be part of the study area. Response: We have revised the manuscript to include a new section that distinctly outlines the methods used for analyzing the landscape structure and configuration in the study area. This additional section specifically describes the analysis of agroforestry systems involving cacao and silvopastoral systems, which were the focus of the study.

L148-149. Could it include the main species that make up the SSP? Does SAF with cocoa present high or low plant diversity? Response: In response to the new revisions and adjustments, we have included a comprehensive description in the Methods section detailing the composition and structure of the agroforestry systems with cacao and the silvopastoral systems sampled in the Colombian Amazon. This description now highlights the key species that constitute each system, providing a clearer understanding of their ecological and functional characteristics.

L170-173. What would be the area of each of the 78 plots? How were the plots within each mosaic systematized? Response: The area of each plot is presented in the abstract of the initial manuscript. In response to the reviewer’s comment, we have added clarification in the Methods section, specifically in the part describing the quantification of biomass in carbon deposits. It is now explicitly stated that each plot covers an area of 0.1 hectares, with dimensions of 20x50 meters. We have also included the relevant citation that suggests the appropriate plot size.

L99-211. This topic is confusing. How was the carbon stored in each plot converted to SAF, SPP, and mosaic model scale? Did the authors use the same equation to convert aerial biomass into root biomass for all vegetation types

---

## [Editor Report · Decision Letter 1]

Influence of Landscape Structure on Carbon Storage in Agroforestry Systems with Cacao and Silvopastoral Systems in the Colombian Amazon

PONE-D-24-52526R1

Dear Dr. Jenniffer Diaz Chaux,

We’re pleased to inform you that your manuscript has been judged scientifically suitable for publication and will be formally accepted for publication once it meets all outstanding technical requirements.

Kind regards,

Marcela Pagano, Ph.D, M.D.

Academic Editor

PLOS ONE
---

## [Editor Report · Acceptance letter]

PONE-D-24-52526R1

PLOS ONE

Dear Dr. Díaz-Cháux,

I'm pleased to inform you that your manuscript has been deemed suitable for publication in PLOS ONE. Congratulations! Your manuscript is now being handed over to our production team.

Kind regards,

on behalf of

Dr. Marcela Pagano

Academic Editor

PLOS ONE